# My Exposed Body: Psychometric Properties of the Italian Version of the Social Physique Anxiety Scale-7 among Women

**DOI:** 10.3390/bs13030224

**Published:** 2023-03-03

**Authors:** Giulia Rosa Policardo, Camilla Matera, Cristian Di Gesto, Amanda Nerini

**Affiliations:** 1Department of Health Sciences, University of Florence, 50135 Florence, Italy; 2Department of Education, Languages, Intercultures, Literatures and Psychology, University of Florence, 50135 Florence, Italy

**Keywords:** social physique anxiety, body dissatisfaction, psychometric validation, invariance, body appreciation, body compassion

## Abstract

Background: Social Physique Anxiety (SPA) is the anxiety resulting from the prospect or presence of the interpersonal evaluation of one’s physique. It is a construct related to body image and body esteem. The Social Physique Anxiety Scale-7 (SPAS-7) is a self-report scale aimed at measuring the degree of anxiety that people experience when others evaluate their physique. Methods: This study aimed to investigate the factor structure (through Confirmatory Factor Analysis followed by multi-group confirmatory factorial analyses), reliability, and convergent validity of an Italian version of the SPAS-7 among a sample of women (*N* = 520; mean age = 33.5, *SD* = 10.5). Results: Confirmatory Factor Analysis attested the unidimensional factorial structure of the SPAS-7, which achieved full invariance across age groups. The strength of the inter-relationships between the SPAS-7 and measures of negative (i.e., body dissatisfaction) and positive body image (i.e., body appreciation and body compassion) provided evidence of good convergent validity. The Cronbach’s alpha was very good. Conclusion: According to our results, the Italian version of the SPAS-7 could be a valid and agile instrument for assessing self-presentational concerns associated with body image among Italian-speaking women across age.

## 1. Introduction

Social Physique Anxiety (SPA) has been defined as a subtype of social anxiety representing the tendency to become apprehensive and anxious in social settings in which one’s body characteristics can potentially become the object of (negative) evaluation by others [1,2]. SPA does not correspond to body dissatisfaction, since it implies not only a negative self-evaluation of one’s general appearance (i.e., body dissatisfaction) [3], but also the emotional component of fear of being judged by others for one’s physical characteristics (e.g., body fat and muscular tone). Whereas SPA is generally conceptualized as a complex and deeply held emotional process [4], body dissatisfaction is described as a cognitive evaluation of one’s body that involves negative affect [5].

Some studies, e.g., [3,6], have shown a strong link between SPA and women’s body dissatisfaction, although they represent two different aspects of negative body image. A negative relationship with one’s appearance is what links the two constructs.

SPA is associated with health-related factors, such as depressive symptoms [7], drive for muscularity and thinness [8], and eating disorders [7]. It was also found to predict positive body image, such as body appreciation, with which it presents a strong negative correlation [9]. Notably, high levels of self-compassion (i.e., an attitude of kindness and acceptance towards oneself) are associated with lower levels of SPA [10], which suggests that having a compassionate attitude towards one’s perceived body inadequacies (i.e., body compassion) could represent a protective factor for anxiety with respect to one’s physical appearance.

The aforementioned evidence about social physique anxiety and its negative consequences on health has been provided through studies on college women and female adolescents. Indeed, women tend to experience significantly higher levels of SPA than men [11]. Negative interpersonal evaluations of one’s appearance seem to be one of the main sources of concerns for women during both adolescence and adulthood [12]. This is particularly true for Western societies, such as Italy, in which the importance of social approval and the desire to appear attractive for potential intimate partners are pervasive [13,14]. Indeed, body dissatisfaction is highly widespread among Italian women [15], who show high levels of anxiety about overall appearance [4]. These factors represent a risk for eating disorders, which affect almost 3 million people in this country with an estimated lifetime prevalence of 1.7% [16]. Efforts in managing how women look, along with the high likelihood of receiving feedback on appearance, create the conditions for social physique anxiety to emerge, both in real and virtual environments [17].

Considering the role that SPA can play on the onset, maintenance, and exacerbation of body-image concerns, it is important to have valid and reliable measures for the consideration and recognition of emotional processes related to Italian women’s body image, so as to intervene early for improving their psychological functioning. While there are validated measures in the Italian context for detecting body dissatisfaction (for example, the Body Shape Questionnaire-14 [18]), there are no validated tools for the measurement of SPA. Dakanalis et al. [4] validated the Italian version of the Social Appearance Anxiety Scale (SAAS) [19], which measures a construct that could be confused with SPA, although it assesses different aspects. Indeed, Social Appearance Anxiety is conceptualized as the fear and anxiety of being judged negatively based on one’s physical appearance in general [19], while SPA refers to the discomfort about negative evaluation by others regarding the form, characteristics, and structure of one’s body [20]. White and Warren [21] showed that these two constructs (namely, Social Appearance Anxiety and Social Physique Anxiety) are distinct; while SPA was found to predict women’s body-checking behaviors and psychosocial functioning, Social Appearance Anxiety was not a significant predictor of these outcome variables. Furthermore, current sociocultural influences on body image, especially Social Network Sites (SNSs), reinforce the importance of managing not so much one’s general appearance, but specific parts of one’s body (e.g., digital modifications focused on improving muscularity, reducing body fat or ameliorating specific body shape); this body-focused attention increases anxiety and distress related to certain aspects of one’s body in adult women [22].

Although many empirical studies have examined body-image concerns, dieting, and disordered eating in young women, evidence of these concerns among adult and aging women continues to accumulate [23]. Women entering midlife begin to experience normal biological age-related body changes, including changes in weight distribution and decreases in skin elasticity or muscle mass [24,25]; consequently, with age, women naturally begin to diverge from the thin-ideal, which is a widespread sociocultural ideal of beauty [26].

It is important to have valid instruments that can measure the specific anxiety that women feel about the probability that specific characteristics of their own bodies can be negatively evaluated by others. These instruments should prove to be valid for differently aged women, given that the tendency to become apprehensive and anxious in social settings in which one’s body characteristics are evaluated can vary across different periods of one’s life [27].

### 1.1. The Social Physique Anxiety Scale

The Social Physique Anxiety Scale (SPAS) was developed by Hart et al. [1] solely to assess perceived anxiety (e.g., nervousness and tension) in situations where one’s body characteristics may be negatively evaluated by others (“There are times when I am bothered by thoughts that other people are evaluating my weight or muscular development negatively”). It can be easily used to guide empirical research aimed at identifying antecedents and effects of body image evaluation. The SPAS was originally developed as a 12-item self-report scale used to assess the degree to which people become anxious when others observe or evaluate their physique (in terms of body proportions, body fat, muscular tone) [1].

However, concerns about its factor structure and some negatively worded items within it prompted some to recommend the deletion of some items from the 12-item version [28]. A further validation study of the SPAS [2] showed that the 12-item version, originally validated by Hart et al. [1], could be reduced to a 7-item scale. This study demonstrated the robustness and reliability (*α* = 0.72) of the 7-item scale while also confirming the unidimensional structure of the SPAS. The 7–item model [2] has provided more evidence of its validity and reliability especially in Western countries, e.g., [29].

In Italy, where the pressures on aesthetic appearance are relevant in real and virtual contexts [15], only one preliminary study [30] has been conducted to investigate, through Exploratory Factor Analysis (EFA), the dimensionality of the Italian version of the SPAS-7, finding good psychometric properties of the 7-item scale compared to other versions (i.e., the 12-item original version). The Cronbach alpha of the SPAS-7 was 0.85. Convergent validity (i.e., the extent to which responses on an instrument exhibit a strong relationship with responses on conceptually similar instruments that are supposed to be related to each other are, in fact, related), as measured by calculation of the correlation between body dissatisfaction and SPA, was good (*r* = 0.58; *p* < 0.001).

The Italian adaptation of the SPAS, although useful for a preliminary analysis of its psychometric properties, has some limitations that we aimed to overcome through the present study. First, only an EFA was performed, which does not enable robust information on the stability of the factorial structure of the SPAS-7. Second, convergent validity was tested only with respect to body dissatisfaction. Third, the age invariance of the scale was not examined.

### 1.2. The Present Study

The present study aimed at confirming the unidimensional structure of the 7-item version of the SPAS among Italian women, using Confirmatory Factor Analysis (CFA). The reliability and construct validity, assessed in terms of convergent validity, of the Italian version of the SPAS-7 were examined as well. In order to provide evidence of convergent validity, SPA was expected to be positively correlated with body dissatisfaction and negatively correlated with body appreciation and body compassion, which can be defined as feeling kindly toward one’s body in the face of perceived defects or inadequacies [31]. Moreover, we expected the unidimensional structure of the 7-item scale to be invariant across women’s age.

## 2. Materials and Methods

### 2.1. Participants and Procedure

Participants were 520 Italian young and adult women aged 25–81 years (*M* = 33.5, *SD* = 10.5). Specifically, 50.6% of participants were under 30 years old (*n* = 263; *M* = 26.3, *SD* = 1.32) and 49.4% were over 30 years old (*n* = 257; *M* = 40.7, *SD* = 10.9).

The mean BMI of the participants was 22.06 (*SD* = 4.10). Most of them (63.5%) lived in central Italy, while 19.2% lived in southern Italy or on islands, and 17.3% lived in northern Italy. More than half of the participants (59.2%) reported being unmarried, 36.5% reported being married or cohabiting, and only 4.2% of the participants were widowed or separated. Regarding education, 34.8% of them had master’s degrees, 30.4% had bachelor’s degrees, 28.7% had high school diplomas, 3.7% had specialized degrees or PhDs, and only 2.5% had middle school licenses. Most of the participants (64.3%) defined themselves as workers, 28.8% were students, 6.3% of them were housewives, and only the 0.3% were retired. We recruited the participants online using virtual social spaces (Facebook, Instagram, and WhatsApp). We invited the participants to take part in a university survey about women’s body image.

Participation was voluntary, and we did not provide incentives to the participants. To be eligible for the study, the women had to be at least 25 years old (in order to reach an adult sample). We obtained online informed consent from each participant, after which they completed an online survey. The survey was anonymous and took about 10 min to complete. All procedures performed in studies involving human participants were in accordance with the 1964 Declaration of Helsinki and its later amendments or comparable ethical standards. To ensure informed consent, in the first page of the online survey, participants were informed about the objectives of the study, procedures of data storage, the voluntary nature of study participation, and their right to withdraw at any time. Individuals were allowed to refuse participation. The study procedures were approved by the local University Ethics Committee (authorization n. 0174350).

### 2.2. Measures

Social Physique Anxiety. In the preliminary study by Nerini et al. [30], the original English version of the SPAS-7 [2] was translated into Italian by two native speakers and then independently back-translated to ensure accuracy. In the present study, we used this preliminary Italian version [30] which consists of 7 items (e.g., “In the presence of others I feel apprehensive about my physique/figure”) rated on a 5-point Likert scale (1 = not at all characteristic of me; 5 = extremely characteristic of me). High scores are related to high social physique anxiety, except for Item 5, which is formulated inversely (e.g., “I feel comfortable about how others appraise my body”).

Body Dissatisfaction. The Italian version [18] of the Body Shape Questionnaire-14 (BSQ-14) was used to assess female body dissatisfaction. The scale has 14 items (e.g., “I felt ashamed of my body”) rated along a 6-point Likert scale (1 = never; 6 = always). High scores indicated greater levels of general body discontentment.

Body Appreciation. The Italian version [32] of the Body Appreciation Scale-2 (BAS-2) was used to assess appreciation towards one’s body. The scale has 13 items (e.g., “I feel good about my body”) rated on a 5-point Likert scale (1 = never; 5 = always). High scores indicated greater levels of body gratefulness.

Body Compassion. The Italian version [33] of the Body Compassion Scale (BCS) was used to assess compassion towards one’s body. The scale has 23 items (e.g., “I am accepting of my looks just the way they are”) rated on a 5-point Likert scale (1 = almost never; 5 = almost always). High scores indicated greater levels of body compassion.

Sociodemographic Details. Participants were asked to indicate their year of birth (to obtain the participant’s age), biological sex, educational level, place of residence, occupational status, and relationship status.

Body Mass Index. Participants were asked to indicate their weights and heights and then we calculated self-reported BMIs (kg/m^2^).

### 2.3. Analysis of Data

For all analyses we used IBM SPSS Statistics 23 and Amos Graphics (EmuLisrel 6).

Preliminary Item Analysis. There were no missing data. Descriptive statistics were examined. The distribution of scores was investigated to identify items with excessive skew (>2) and kurtosis (>7) values [34]. Factor loadings values were considered appropriate if equal to or greater than 0.50 according to Costello and Osborne [35].

Fit and reliability indices. A CFA was performed to examine the fit of the 7-item model. We used the Satorra–Bentler Scaled Chi-Square (S–B χ^2^), the Comparative Fit Index (CFI), the Tucker Lewis Index (TLI), the Root Mean Square Error of Approximation (RMSEA), and the Standardized Root Mean Square Residual (SRMR) as principal indices of goodness of fit. We considered CFI and TLI values between 0.90 and 0.95 as suggesting reasonable fit, and RMSEA and SRMR values of 0.05 and 0.08 as indicating good and moderate fit, respectively [36]. Cronbach’s alpha was used to test the internal consistency of the Italian version of the SPAS-7. Values in the interval 0.70–0.90 were considered optimal, while values above 0.60 were deemed acceptable [37].

Convergent validity. We assessed convergent validity via bivariate Pearson correlation coefficients between the SPAS-7 and other measures of negative (body dissatisfaction) and positive body image (body compassion and body appreciation). Based on the literature, we hypothesized that high levels of SPA would positively correlate with high levels of body dissatisfaction [6,38]. Similarly, we expected to find strong negative correlations between SPA and constructs that assess positive body image (body appreciation and body compassion). A correlation of 0.10 was considered small, 0.30 was considered medium, and 0.50 or more was considered large [39].

Invariance across age. Multi-group CFAs were conducted to test the invariance of the SPAS-7 across age by using the “step-down” methodology [40]. The sample was divided into two groups: one group consisted of women aged under 30 (*n* = 263; *M* = 26.3, *SD* = 1.32) and the second group consisted of women aged over 30 (*n* = 257; *M* = 40.7, *SD* = 10.9). According to Byrne [40], the testing strategy begins by testing the confirmatory factor analysis within each group separately for evidence of fit. Assuming a reasonably good fitting model (see the “fit and reliability indices” paragraph for cut-off criteria), one proceeds to test for evidence of configural invariance (also referred to as form invariance), metric invariance (also referred to as weak invariance), and scalar invariance (also referred to as intercept invariance).

Sample Size and Power. For sample size and statistical power, we followed the widely accepted ratio of 10:1 (cases per item) for CFA [41]. According to widely accepted guidelines, the statistical power of the present study was optimal, as we had more than 50 participants per item.

## 3. Results

Table 1 shows descriptive statistics and correlations of all items that compose the Italian version of the SPAS-7. Skewness was lower than 2 and kurtosis was lower than 7 for all of the items. Women’s scores were close to the middle-point of the scale (*M* = 2.94). All the items’ intercorrelation were positive, except for item 5 that is drafted inversely.

The model presented a very good fit to the data [X^2^/gdl = 4.48; *p* = 0.001; CFI = 0.98; TLI = 0.96; SRMR = 0.03; RMSEA = 0.08 (Confidence Interval = 0.06; 0.10)].

Factor loadings are presented in Figure 1. All of them were greater than 0.50, ranging from 0.59 to 0.85.

Table 2 shows the diverse fit indexes for the three models compared in the invariance analyses across age. No significant differences were found among the configural, metric, and scalar invariance models. The lack of significant differences between the first two models entails a minimal criterion for accepting the existence of the model’s invariance across age [42].

The internal consistency of the SPAS-7 with the total sample was excellent (α = 0.90). Furthermore, the Cronbach’s alpha for the two subsamples was excellent (for women older than 30: α = 0.91; for women under the age of 30: α = 0.89).

Descriptive statistics and Cronbach’s alpha of the measures used for convergent validity are displayed in Table 3.

Table 4 summarizes the intercorrelations between the SPAS-7, BSQ-14, BAS-2, and BCS. The SPAS-7 presented strong positive association with BSQ-14. Moreover, we found strong and negative associations between the SPAS-7 and both the BAS-2 and the BCS.

## 4. Discussion

The results confirmed the unidimensional structure of the Italian version of the SPAS-7 and its good psychometric properties with Italian women across age. The SPAS-7 can be used in the Italian context for evaluating the tendency to become apprehensive and nervous in social situations in which one’s body or body shapes can potentially become the object of observation and evaluation by others. In line with recent research [27], the 7-item unidimensional structure of the SPAS was invariant across age.

The reliability for the Italian version of the scale was supported by optimal levels of internal consistency (α = 0.90). This value was higher than the one found for the original English version (α = 0.72) and the preliminary Italian version (α = 0.85) [30].

The convergent validity of the Italian version was good too. Specifically, the SPAS-7 showed a positive and strong correlation with women’s body dissatisfaction. Higher feelings of distress given by the expectation that one’s physical self is subject to social evaluation was associated with higher concerns and negative thoughts about one’s body. These results are in line with an experimental study showing that high social physique anxiety, measured by a preliminary Italian version of the SPAS-7 [30], was strongly associated with higher body dissatisfaction in young women [17].

The negative association between positive body image (namely body appreciation and body compassion) and social physique anxiety also suggested a good convergent validity of the SPAS-7. These results were in line with a recent experimental study [9], in which higher body appreciation was associated with lower SPA. Our findings suggested also that women with a higher attitude of kindness and acceptance of perceived limits and inadequacies in relation to one’s body (i.e., body compassion) may present a lesser tendency to become worried and apprehensive about evaluating one’s physical aspects in social contexts, which is partially in line with the study by Koç and Ermis [10], who found that higher self-compassion levels were associated with lower levels of SPA.

In sum, given its low number of items and good psychometric properties, the SPAS-7 represents a valuable instrument for measuring social physique anxiety among Italian women. Nevertheless, the present study has some limitations that should be considered. The reliability of the SPAS-7 was assessed only in terms of internal consistency. It would be interesting to test other forms of reliability, such as test-retest, through longitudinal research designs that could examine the reliability of the scale over time. It might also be interesting to assess the predictive validity of the SPAS by positing this variable as a predictor of physical self-esteem or, more generally, of women’s social/interpersonal quality of life and wellbeing. Furthermore, in the present study, we did not measure if participants had body image concerns (e.g., eating disorders or dysmorphic disorders) or other mental problems such as the presence of Social Anxiety Disorder. Future studies could collect this kind of clinical information because these aspects could affect the results obtained in the SPAS-7. Finally, it should be noted that online data collection may suffer from self-selection bias (i.e., only people who are present on Social Media and know how to use it can participate).

In this study we considered a female sample because of the prevalence and incidence of SPA among women across age [43]. Women are more likely to compare their physique to others, particularly in relation to their weight and attractiveness, which triggers higher levels of SPA [44]. However, concerns about physical appearance in terms of body parts and proportions are also increasing among the Italian male population, e.g., [45,46]. Further examination of this specific process (fear of being judged for one’s physical characteristics) among Italian men represents an important area of research. Therefore, it would be important to test the validity and sensitivity of the SPAS-7 also in the Italian male population.

## 5. Conclusions

Our findings contribute to the body of research on body image assessment with a particular focus on the salience of social environments. The Italian version of the SPAS-7 seems to be useful for measuring an important aspect of women’s self-presentational concern that, in the present study, was associated with issues related to negative (i.e., body dissatisfaction) and positive body image (i.e., body appreciation and body compassion). This scale could be useful to plan health promotion interventions aimed at regulating social physique anxiety responses, thus improving the relationship that Italian women have with their physical selves. To assess the individual’s social physique anxiety with just a few items (see Appendix A) proves advantageous in the execution of large-scale studies, as the time spent and effort made by participants are reduced [47].

Consequently, the SPAS-7 could be a very useful measure that could be easily and quickly applied in various settings (e.g., clinical settings), given the small number of items that compose it. Future studies using the SPAS-7 for detecting social physique anxiety will also provide new insights in the field of general social anxiety, given that appearance-focused concerns may be included as one of the concerns and treatment targets regarding the Social Anxiety Disorders spectrum [4].

## Figures and Tables

**Figure 1 behavsci-13-00224-f001:**
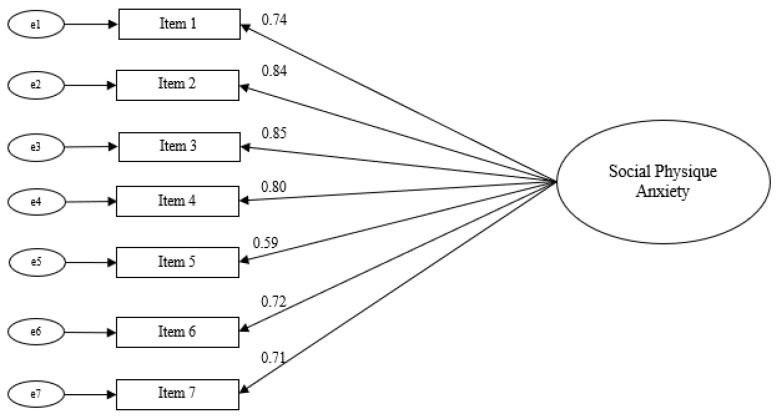
Confirmatory Factor Analysis Model and factor loading (*N* = 520).

**Table 1 behavsci-13-00224-t001:** Descriptive Statistics and correlations of all the 7 items of the SPAS-7 (*N* = 520).

	Mean (*SD*)	Skewness (*SD*)	Kurtosis (*SD*)	1	2	3	4	5	6	7
Item 1	2.75 (1.26)	0.31 (0.11)	−0.93 (0.21)	-	0.63 ***	0.64 ***	0.61 ***	−0.48 ***	0.50 ***	0.45 ***
Item 2	2.64 (1.35)	0.33 (0.11)	−1.13 (0.21)		-	0.70 ***	0.65 ***	−0.48 ***	0.66 ***	0.62 ***
Item 3	2.72 (1.29)	0.22 (0.11)	−1.10 (0.21)			-	0.72 ***	−0.53 ***	0.58 ***	0.60 ***
Item 4	2.34 (1.21)	0.62 (0.11)	−0.59 (0.21)				-	−0.49 ***	0.54 ***	0.56 ***
Item 5 *	2.65 (1.13)	0.13 (0.11)	−0.86 (0.21)					-	−0.37 ***	−0.40 ***
Item 6	3.11 (1.28)	−0.11 (0.11)	−1.13 (0.21)						-	0.59 ***
Item 7	2.96 (1.33)	0.08 (0.11)	−1.21 (0.21)							-
SPAS-7 (total)	2.94 (1.00)	-	-							

Note: Likert-scale answer to 5 points from 1 to 5. * Reversed item. *** *p < 0*.001.

**Table 2 behavsci-13-00224-t002:** Results of the CFAs testing the factorial invariance of the unidimensional model across age.

Model	X^2^_(df)_	CFI	TLI	RMSEA [90% CI]	ΔX^2^	Δ_df_	*p*	Δ_cfi_
Configural Invariance	78,599_(28)_	0.974	0.962	0.06 [0.04–0.07]	-	-	-	-
Metric Invariance	84,466_(34)_	0.973	0.967	0.05 [0.04–0.07]	5.876	6	0.43	0.001
Scalar Invariance	88,577_(40)_	0.975	0.974	0.05 [0.03–0.06]	4.111	6	0.66	0.002

Note: df = degrees of freedom; CFI = Comparative Fit Index; TLI = Tucker–Lewis Index; RMSEA = Root Mean Square Error of Approximation; ΔX^2^= X^2^ scaled difference; Δ_df_ = difference in degree of freedom between nested models; *p* = probability values of R^2^ test; Δ_cfi_ = difference in CFIs between nested models.

**Table 3 behavsci-13-00224-t003:** Descriptive statistics and Cronbach’s alpha of the Social Physique Anxiety, Body dissatisfaction, and Body Compassion (*N* = 520).

	Min	Max	Mean (*SD*)	*α*
SPAS-7	1	5	2.94 (1.00)	0.90
BSQ-14	1	5.93	2.97 (1.13)	0.95
BAS-2	1	5	3.33 (0.80)	0.94
BCS	1	5	3.16 (0.80)	0.93

Note: Social Physique Anxiety Scale-7 (SPAS-7); Body Shape Questionnaire-14 (BSQ-14); Body Appreciation Scale (BAS-2); Body Compassion Scale (BCS).

**Table 4 behavsci-13-00224-t004:** Bivariate Correlations Between Social Physique Anxiety, Body dissatisfaction, Body Appreciation, and Body Compassion (*N* = 520).

	1	2	3	4
1. SPAS-7	-			
2. BSQ-14	0.77 ***	-	-	
3. BAS-2	−0.69 ***	−0.62 ***	-	-
4. BCS	−0.66 ***	−0.67 ***	−0.70 ***	-

Note: Social Physique Anxiety Scale-7 (SPAS-7); Body Shape Questionnaire-14 (BSQ-14); Body Appreciation Scale-2 (BAS-2); Body Compassion Scale (BCS). *** *p* < 0.001.

## Data Availability

The data presented in this study are available on request from the corresponding author. The data are not publicly available due to ethical, local reasons.

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
