# Peer review of "My Exposed Body: Psychometric Properties of the Italian Version of the Social Physique Anxiety Scale-7 among Women"

_behavsci, 2023, doi:10.3390/bs13030224_

Round 1

Reviewer 1 Report

It is a great pleasure to review this manuscript. The authors examined the psychometric qualities of the Italian version of the Social Physique Anxiety Scale-7 (SPAS-7). Confirmatory factor analysis supported the unidimensional structure suggested by the exploratory factor analysis result of a past study. Moreover, the SPAS-7 also showed good internal consistency and satisfactory (convergent and divergent) validity.

The manuscript is well written. It is enjoyable to read it. Below are the comments to help the authors to further enhance the quality of the manuscript.

Main concern

The examination of validity requires clarification. In the Analysis of data section, please explicitly indicate the variables used to test the two types of validity (e.g., BMI used for testing convergent validity). More importantly, justify using the variables. On lines 29 to 34, the authors reported that social physique anxiety (SPA) does not correspond to body dissatisfaction, implying that they are two different concepts. In that case, it is unclear why body dissatisfaction is suitable to test the convergent validity of the SPAS-7. Moreover, the authors also wrote that “SPA positively predicts women’s body dissatisfaction”, suggesting that body dissatisfaction is a potential variable for testing the criterion-related validity of the SPAS-7. The same inquiry also applies to the use of BMI.

Similarly, the rationale for examining the hypothetical negative relationship between the SPAS-7 score and body appreciation to test divergent validity requires further explanation. The APA Dictionary of Psychology defines divergent validity as “the degree to which a test or measure diverges from (i.e., does not correlate with) another measure whose underlying construct is conceptually unrelated to it.”  Hence, it is unclear how the strong negative relationship between SPA and body appreciation leads to the conclusion that the divergent validity of the SPAS-7 is supported.

The authors shall also clarify the relationship between SPA, body dissatisfaction, and body satisfaction. SPA was found to have a strong positive relationship with body dissatisfaction and a strong negative relationship with body appreciation, respectively. Meanwhile, body dissatisfaction and body appreciation were also highly (and negatively) correlated. The results seem to indicate that SPA is conceptually similar to the perception of negative body image.   

Minor issues

1.      Provide examples of the physique in the first paragraph to help readers distinguish physique from body dissatisfaction and physical appearance.

2.      Line 65, insert a citation for the Social Appearance Anxiety Scale.

3.      Line 104, further elaborates on the meaning of “good psychometric properties”. The authors are suggested to report the findings of the past study (e.g., reliability) and then compare them with the findings of the present study in the Discussion.

4.      Although women are prone to SPA, it is also equally important to investigate whether the SPAS-7 is useful for men. Please clarify the reason the present study did not include male participants.

5.      Line 120, the sample was composed of young adult women, but the highest value of age was 81. Please verify it.

6.      TLI and SRMR and their suggested cut-off were not mentioned in Section 2.3 Analysis of data.

7.      Define the abbreviation of CI.

8.      The discussion of the limitations of the present study can be further enriched. For example, the predictive validity of the SPAS-7 shall be tested.

9.      Minor language and spelling errors are detected.

Reviewer 2 Report

Dear Authors,

Thank you for your manuscript. Please see my comments below.

Abstract. Please indicate the mean (± SD) and range age of the study participants. Also, please revise these sentences: "Confirmatory Factor Analysis confirmed the construct validity of the SPAS-7 among Italian women". Did you mean the unidimensional structure of the scale was confirmed? "The reliability of the scale was very good". Did you mean Cronbach's alpha was good?

Methods. Was the translation procedure of the SPAS-7 described elsewhere?

In section 2.3. references supporting important statements should be provided:

"Item distribution was investigated to identify items with excessive skew (> 2) and kurtosis (> 7) values".

"Cronbach’s alpha was used to test the internal consistency reliability of the Italian version of the SPAS-7. Values in the interval .70 – .90 are considered optimal, values above .60 are acceptable as well".

"A correlation of .10 was considered small, .30 medium and .50 183 or more was considered large".

Finally, have the authors calculated the estimated power for this study?

Also, the MDPI does not support APA styling in numbers: .70 should be 0.70, etc.

As a major limitation, I see the age of the study participants and selection procedures. The age range is very broad (25-81). Most of the scales associated with body image concerns are designed to reflect young individuals' perceptions. It would be helpful to provide the % of the study participants aged ≤ 30 and >30 years. Also, could the study participants be self-selective if they were recruited from social media (FB, Instagram) and possibly are completely different according to body image concerns compared to the general population?

Results. In Table 1, it would be helpful to provide the original English statements together with the Italian.

In Tables 2 and 3, please correct BAS into BAS-2.

Also, the authors do not perform multigroup analysis for invariance testing. Taking into account that this is not the first attempt to validate SPAS, the analysis is incomplete, again.

Please address these essential limitations.

Round 2

Reviewer 1 Report

I thank the authors for their efforts in addressing my comments and improving the quality of the manuscript.

I have two more comments for the authors:

1. The rationale for using BMI in testing the convergent validity of the SPAS-7 does not convince me. It is agreed that BMI is associated with body image concerns and fear of being judged in a negative way because individuals with high BMI tend to be more concerned about their body image than those with low BMI. However, this relationship does not mean that BMI and body image concerns are conceptually similar. Hence, it makes little sense to test convergent validity using BMI. I'm grateful if the authors can provide further explanations to help us understand the rationale.   

2. Report the cut-off criteria for testing measurement invariance used in the present study. 

Reviewer 2 Report

Dear Authors,

Thank you for taking into account my concerns and remarks. I appreciate it. Good job has been done. All the best!
